# Molecular Epidemiology of Fosfomycin Resistant *E. coli* from a Pigeon Farm in China

**DOI:** 10.3390/antibiotics10070777

**Published:** 2021-06-25

**Authors:** Lu Han, Xiao-Qing Lu, Xu-Wei Liu, Mei-Na Liao, Ruan-Yang Sun, Yao Xie, Xiao-Ping Liao, Ya-Hong Liu, Jian Sun, Rong-Min Zhang

**Affiliations:** 1National Risk Assessment Laboratory for Antimicrobial Resistance of Animal Original Bacteria, College of Veterinary Medicine, South China Agricultural University, Guangzhou 510642, China; luhan@stu.scau.edu.cn (L.H.); lxq@stu.scau.edu.cn (X.-Q.L.); xwliu11@stu.scau.edu.cn (X.-W.L.); lmn@stu.scau.edu.cn (M.-N.L.); sunruanyang@163.com (R.-Y.S.); 2017zrm@stu.scau.edu.cn (Y.X.); xpliao@scau.edu.cn (X.-P.L.); lyh@scau.edu.cn (Y.-H.L.); jiansun@scau.edu.cn (J.S.); 2Guangdong Provincial Key Laboratory of Veterinary Pharmaceutics Development and Safety Evaluation, College of Veterinary Medicine, South China Agricultural University, Guangzhou 510642, China; 3Guangdong Laboratory for Lingnan Modern Agriculture, College of Veterinary Medicine, South China Agricultural University, Guangzhou 510642, China

**Keywords:** fosfomycin resistance, pigeon farm, *E. coli*, *fosA3*, transmission

## Abstract

We determined the prevalence and molecular characteristics of fosfomycin-resistant *Escherichia coli* from a domestic pigeon farm. A total of 79 samples collected from pigeons and their surrounding environments were screened for the presence of fosfomycin resistant isolates and these included 49 *E. coli* isolates that displayed high-level resistance (MIC ≥ 256 mg L^−1^) and carried the *fosA3* gene on plasmids with sizes ranging from 80 to 370 kb. MLST analysis of these *fosA3*-positive *E. coli* isolates indicated the presence of nine sequence types (ST6856, ST8804, ST457, ST746, ST533, ST165, ST2614, ST362 and ST8805) of which ST6856 was the most prevalent (24.5%, 12/49). PFGE combined with genomic context comparative analyses indicated that the *fosA3* gene was spread by horizontal transfer as well as via clonal transmission between *E. coli* in the pigeon farm, and IS*26* played an important role in *fosA3* transmission. The high prevalence of *fosA3* in the pigeon farm and the high similarity of the *fosA3* genomic environment between *E. coli* isolates from humans and pigeons indicated that the pigeon farm served as a potential reservoir for human infections. The pigeon farm was found to be an important reservoir for the *fosA3* gene and this should be further monitored.

## 1. Introduction

The prevalence of bacterial resistance to multiple antibiotics has resulted in fewer treatment options for clinicians. Fosfomycin has emerged as a candidate antibiotic to treat infections caused by carbapenem-, tigecycline-, and polymyxin-resistant bacteria [1]. Resistance to fosfomycin is mediated by FosA enzymes in Gram-negative bacteria and there are more than 10 *fos* types, of which *fosA* and *fosC2* are the primary ones found in the Enterobacteriaceae [2]. The *fosA3* gene has also been co-localized via IS*26* with *bla*_CTX-M_ on epidemic plasmids frequently detected in *E. coli* that spread among humans, pets, and food animals [3,4]. However, fosfomycin is not approved for use in animals in China, although the *fosA3* gene has been detected in food animals as well as humans [5]. 

Pigeon breeding is popular and common in China, especially in Guangdong province, and domestic pigeons are found in all urban areas, with a population size estimated at several hundred million [6]. Moreover, the number of pigeons consumed in China has grown to three times as many as one decade ago [7]. Importantly, clinically healthy homing pigeons have served as an unnoticed reservoir for zoonotic bacteria that are spread by faeces [8]. Furthermore, the close relationships between humans and birds have contributed to the spread of infectious agents, including antibiotic-resistant bacteria [9], and are an epidemiological link to wild birds [8]. The global and clonal multidrug-resistant carbapenem-resistant Enterobacteriaceae (CRE) carried by pigeons therefore may be established in ecological niches that include humans, livestock, wildlife, pets, and insects [10]. 

The goal of the present work was to reveal the distribution and prevalence of fosfomycin-resistant *E. coli* isolates in domestic pigeons. Our results indicated that pigeons carrying *fosA3*-positive *Escherichia coli* were prevalent in pigeon farms and that *E. coli* of human origin are a potential reservoir for the *fosA3* gene.

## 2. Materials and Methods

### 2.1. Bacterial Isolation, Fosfomycin-Resistant Gene Identification, and Antimicrobial Susceptibility Testing

A total of 79 samples including pigeon faeces (*n* = 34), pigeon ceca (*n* = 12), pigeon pericardial fluid (*n* = 6), sewage (*n* = 18), dust (*n* = 3), and flies (*n* = 6) were collected from a pigeon farm in South China, Foshan City, Guangdong province, in 2016. Briefly, flies, sewage, and dust were collected as reported previously [11]. Samples were incubated in Luria–Bertani (LB) liquid medium overnight and the fosfomycin-resistant isolates were selected on MacConkey agar containing 256 mg/L fosfomycin. All colonies were selected and identified using Gram staining and sequence analysis of the 16S rDNA gene using previously described primers [12]. Species identification of Gram-negative bacteria was further conducted by MALDI-TOF MS [13]. Fosfomycin resistance genes, including *fosA*, *fosA3*, *fosC2*, and *bla*_CTX-M_, were screened using PCR and sequencing [14]. Multilocus sequence typing (MLST) for *E. coli* was based on seven housekeeping genes, as reported previously [15]. PCR amplification and sequencing were performed following the protocols suggested on the website [16]. 

### 2.2. Antimicrobial Susceptibility Testing

The minimum inhibitory concentrations (MICs) of a range of antibiotics against the fosfomycin-resistant isolates were determined using the agar dilution method according to the Clinical and Laboratory Standards Institute (CLSI) document M100-S28 [17], and the breakpoints of colistin and tigecycline for Enterobacteriaceae were interpreted according to EUCAST criteria (Version 7.0) [18]. The *E. coli* isolate ATCC 25922 was used for quality control.

### 2.3. Molecular Analysis of Fosfomycin-Resistant E. coli

The fosfomycin-resistant *E. coli* were investigated by *Xba*I pulsed field electrophoresis (PFGE) and the locations of the fosfomycin resistance genes in the original isolates were determined by S1-PFGE mapping and Southern blot analysis, as described previously [19]. The PFGE patterns were analysed using a dendrogram of per cent similarity calculated with Dice coefficients using a cut-off of 80% [20]. The transferability of fosfomycin resistance genes was determined by the filter mating method using streptomycin-resistant *E. coli* C600 as a recipient strain, in accordance with a previous study [19].

### 2.4. Whole-Genome Sequencing and Phylogenetic Analysis of fosA3-Positive Isolates

Total DNA was extracted using a DNA Extraction Kit (TianGen, Beijing, China) and subjected to whole-genome sequencing (WGS). The library was constructed using the NEXT Ultra DNA Library Prep Kit (New England Biolabs, Ipswich, UK) and an Illumina HiSeq 2500 system (Novogene, Guangzhou, China), which produced 250 bp paired-end reads. A draft assembly of the sequences was generated using CLC Genomics Workbench 5 (CLC Bio, Aarhus, Denmark). All contigs were searched for *fosA3* genes using standalone BLAST analysis [21].

To explore the molecular relationships among the isolates in the current study and other resources, WGS information for 130 *E. coli* isolates was downloaded from GenBank as of 7 November 2020. (Appendix A). The 130 representative isolates were selected according to the following standard. Briefly, the WGS of a total of 774,435 isolates were downloaded from GenBank and the *fosA3*-positive *E. coli* were screened using stand-alone BLAST analysis, resulting in a total of 915 *fosA3*-positive *E. coli*. Phylogenetic analysis was performed using the isolates from the current study and the public repository. Since the phylogenetic software could only process 500 isolates at a time, these 915 *fosA3*-positive *E. coli* strains were randomly divided into two groups and each group was employed to construct a phylogenetic tree with the 14 isolates in the current study, respectively. Then, the 130 representative strains were selected according to their distribution in the phylogenetic tree, sources, and sampling locations (Appendix A). MLST results were analysed using MLST Version 2 [22].

The downloaded genomic sequences, along with time and geospatial information, were applied to construct phylogenetic trees with the *E. coli* isolates using Parsnp in Harvest package 5 [23]. SNP assignments were determined using Snippy v4.4.5 [24] and Snippy-core was used to determine the core SNPs. For the phylogenetic tree, a reference genome was randomly selected using the ”-r!” switch. The *E. coli* population structure was estimated with the hierBAPS module from the software BAPS v6.0 [25], which fits lineage to genome data using nested clustering. Annotations for each isolate and tree embellishment were visualized using iTOL [26].

### 2.5. Data Availability

Genome assemblies of 14 strains co-harbouring *fosA3* were deposited in GenBank under BioProject accession number PRJNA728561.

### 2.6. Ethical Statement

This research was conducted in accordance with the regulations of the ethical guidelines of South China Agricultural University. Written informed consent was obtained from the owners for the participation of pigeons. 

## 3. Results

### 3.1. Identification of fos Genes in Samples

We identified a total of 69 *E. coli* isolates in our 79 samples collected on the pigeon farm that were resistant to fosfomycin (MIC ≥ 256 mg L^−1^), and these included isolates from faeces (*n* = 34), ceca (*n* = 12), liquor pericardii (*n* = 5), sewage (*n* = 14), dust (*n* = 1) and flies (*n* = 3). These isolates were further identified by 16S rDNA gene sequencing combined with MALDI-TOF MS as *E. coli* (*n* = 52), *Proteus mirabilis* (*n* = 15) and *Citrobacter freundii* (*n* = 2). PCR results indicated that the prevalence of *fosA3* in the *E. coli* was 94.2% (49/52) and *fosA* and *fosC* variants were not detected. MLST analysis of these *fosA3*-positive *E. coli* indicated nine types and ST6856 was the most prevalent sequence type (ST) (24.5%, 12/49), followed by ST8804 (18.4%, 9/49), ST457 (16.3%, 8/49), ST746 (12.2%, 6/49), ST533 (10.2%, 5/49), ST165 (8.2%, 4/49), ST2614 (4.1%, 2/49), ST362 (4.1%, 2/49) and ST8805 (2.0%, 1/49). 

The 49 *fosA3*-positive *E. coli* isolates were resistant to ampicillin, cefotaxime, fosfomycin, tetracycline and ciprofloxacin (100%, 49/49), colistin (95.9%, 47/49), trimethoprim/sulfamethoxazole (73.47%, 36/49), florfenicol (34.69%, 17/49), gentamicin (18.37%, 9/49) and amikacin (2.04%, 1/49). All *E. coli* isolates were susceptible to chloramphenicol, meropenem and tigecycline and the fosfomycin-resistant *E. coli* remained highly susceptible to meropenem and tigecycline.

Conjugation experiments were performed on 49 *fosA3*-positive *E. coli* isolates and generated 31 transconjugants (63.27%, 31/49). For the 49 *fosA3*-positive *E. coli* isolates, PFGE was successfully performed and generated 13 pulsotypes designated as pulsotypes A–M. Pulsotype C, which was derived from faeces and sewage, as well as pulsotype I, from faeces, sewage and ceca, were the most prevalent clonal isolates and accounted for 18.4% of the total. Identical PFGE patterns were found between pericardial effusion and flies for pulsotype B as well as from faeces and sewage in pulsotypes E, G, I and K. Pulsotypes K–M were identified as ST457 and pulsotypes H–J as ST6856, whereas each PFGE pattern in the remaining pulsotypes was identified as one type of ST (Figure 1). Among these 49 isolates, S1-nuclease PFGE mapping and Southern blot analysis were performed successfully in 31 (63.2%, 31/49) of them and revealed that the *fosA3* gene was present on differently sized plasmids, including 150 kb (*n* = 10), 80 kb (*n* = 5), 210 kb (*n* = 7), 240 kb (*n* = 1), 310 kb (*n* = 5), 340 kb (*n* = 1) and 370 kb (*n* = 2) (Appendix A).

### 3.2. Whole-Genome Sequencing and Phylogenetic Analysis of fosA3-Positive E. coli Isolates

We performed WGS on two isolates each for pulsotypes C and I, which possessed the largest numbers of isolates, and for single representatives from each of the remaining pulsotypes; only one failed to be sequenced. A total of 130 representative strains with time and geospatial information were then combined with these 14 isolates and a phylogenetic tree was constructed [27]. These 144 isolates were primarily distributed in China (*n* = 126), followed by South America (*n* = 7) and other counties (*n* = 11), and were derived from animals (*n* = 57), humans (*n* = 71) and environmental sources (*n* = 16) (Figure 2). These isolates were further clustered into eight groups sharing a total 163,219 SNPs. MLST analysis revealed that *fosA3* was distributed across a diverse range of STs (*n* = 71). Cluster 2 contained only a single isolate (ST245) and cluster 6 possessed five isolates (all ST457). However, all other clusters contained >2 STs, such as cluster 1 which was dominated by ST167 (14/144), while ST167 (6/144) and ST156 (14/144) were prevalent in cluster 8. Additionally, the strain E7-1 from pigeon faeces shared only 2 SNPs with the strain W1-1 from a sewage isolate in cluster 8 and the strain GZEC065 from human blood shared only 18 SNPs with the strain WFA04 from pig lungs. A strain from a chicken cecum shared only one SNP with strain 648 from a dog anal swab sample (Figure 2).

We identified the presence of 40 antibiotic resistance genes (ARGs) which mediated resistance to 11 types of antibiotics that co-existed with *fosA3* in these 144 *E. coli* isolates. These included resistance genes to fosfomycin, β-lactams, colistin, tetracycline, aminoglycosides, chloramphenicol, quinolones, macrolides, sulphonamides, trimethoprim, and rifampicin (antibiotic resistance gene class is represented with the line colours at the bottom of Figure 2). The β-lactamase genes displayed the greatest diversity and *bla*_CTX-M_ (91%, 131/144) and *mdfA* (98.6%, 142/144) were most prevalent. The genes *bla*_NDM_, *mcr* and *tet*(X4), which mediate carbapenem, colistin, and tigecycline resistance, were present among these *fosA3*-carrying *E. coli* at 38.9, 77.8 and 0.69%, respectively.

### 3.3. Genetic Environments

A comparative genomic analysis indicated that *fosA3* was present in three genomic contexts including types Ⅰ, Ⅲ and Ⅴ, with Ⅰ being the most prevalent, which originated from faeces, ceca and sewage. The entries from the *fosA3*-positive *E. coli* from the public repository indicated that three genomic environments flanking *fosA3* in *E. coli* of human origin resembled those found in current study (types Ⅱ, Ⅳ and Ⅵ). Type III of pigeon origin and type IV of human origin shared the greatest similarities in the backbones and differed only by one IS*26* element, which was complete in type IV but truncated in type III. In the remaining four genomic contexts, IS*26* was present both upstream and downstream of the *fosA3* gene. In addition, the extended spectrum β-lactamase (ESBL) gene *bla*_CTX-M_ was frequently located upstream of the *fosA3* gene in these six genomic contexts (Figure 3). In addition, combining the NCBI database information (998 *fosA3*-positive *E. coli* isolates downloaded on 7 November 2020) with the WGS (14 *fosA3*-positive *E. coli* isolates) analysis of this study, we found that *bla*_CTX-M_ was present in 92.59% (937/1012) of the isolates. The prevalence of *bla*_CTX-M-14_, *bla*_CTX-M-15_, *bla*_CTX-M-55_ and *bla*_CTX-M-65_ was 27.17, 4.94, 36.67 and 17.292%, respectively.

## 4. Discussion

*E. coli* resistant to fosfomycin have become a serious global health problem [28,29]. The high level (71%, 49/69) of *fosA3*-positive *E. coli* detected from pigeons in the current study was significantly higher than that found for pets (10.34%, 3/29) and food animals (1.1%, 10/892) in previous studies for Guangzhou, China [30,31,32]. PFGE patterns combined with the genomic contexts of regions flanking *fosA3* indicated that *fosA3* could spread by horizontal transfer as well as via clonal dissemination between *E. coli* on the pigeon farm, and this most likely led to its high prevalence.

Our isolates also frequently contained IS*26* both upstream and downstream of *fosA3* and this has been previously reported [33]. This is further evidence that IS*26* plays a role in the dissemination of *fosA3* in *E. coli*. IS*26* also facilitated ARG diversity in our isolates, including *aphA1*, *tet*(C), *tet*(D), *catA2* and *cfr* [34]. The *fosA3* gene was generally found co-localized with *bla*_CTX-M_ [34] and this most likely was the result of the combined administration of fosfomycin and β-lactam antibiotics that promoted their co-transfer between Gram-negative bacteria.

The use of fosfomycin in food-producing animals has not been approved in China. Nevertheless, we found a high prevalence of *fosA3* in food-producing animals and in *E. coli*, *P. mirabilis*, *E. fergusonii* and *C. freundii* isolates from pets, and pet owners [4]. The isolates sharing <35 SNPs were classified as clonally spread according to criteria defined in a previous study [35,36]. The phylogeny for a strain from human blood (accession: SAMN13875146) that shared only 18 SNPs with strain WFA04 from a pig suggested that the *fosA3*-carrying *E. coli* were likely spread between humans and food animals via clonal propagation. In addition, we found that the genomic contexts that flanked *fosA3* in the current study closely resembled those from human isolates, suggesting that the pigeon farm served as a potential reservoir for human infections.

## 5. Conclusions

In current study, we detected a high prevalence of *fosA3* in pigeon farm samples and found that *fosA3* could spread by both horizontal and clonal transfer. The high similarity of the *fosA3* genomic environment between *E. coli* isolates from humans and pigeons indicated that the pigeon farm served as a reservoir for these strains. The presence of *fosA3* on the pigeon farm poses a threat to public health and should be regularly monitored.

## Figures and Tables

**Figure 1 antibiotics-10-00777-f001:**
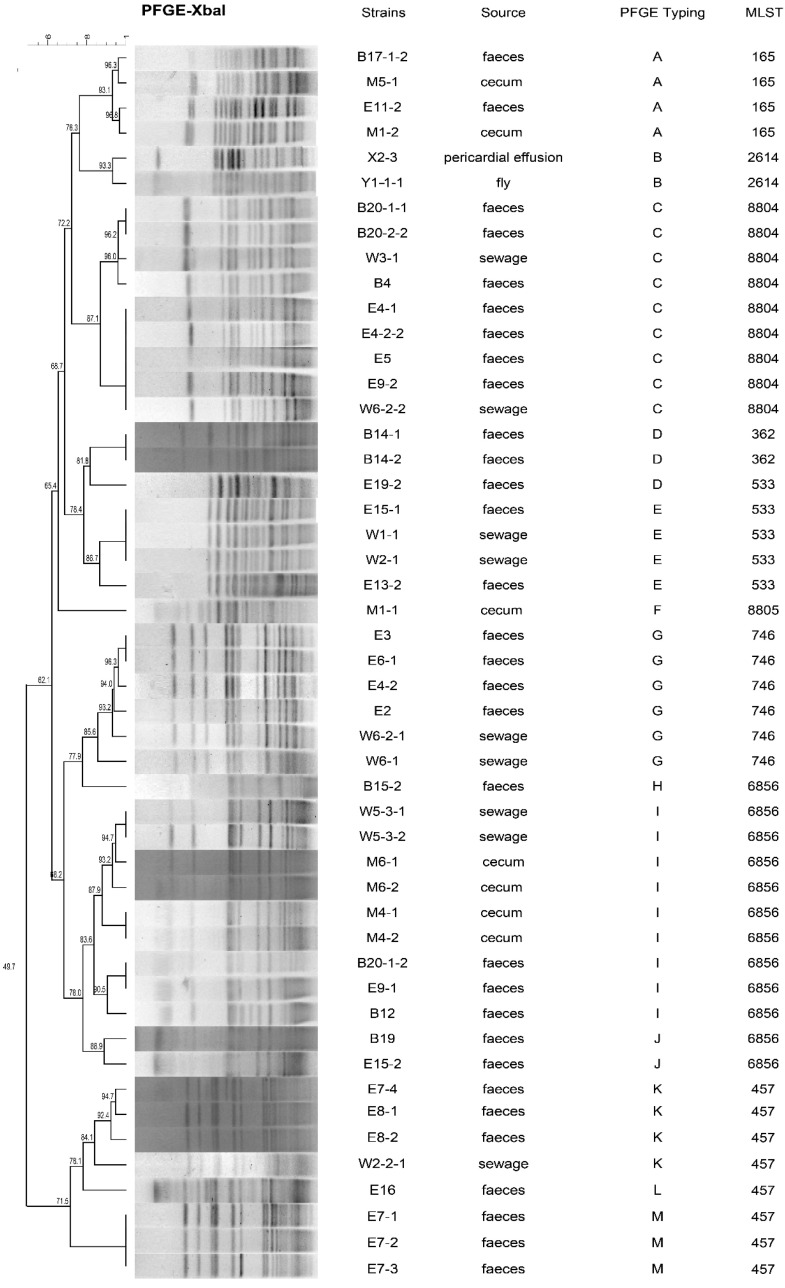
PFGE typing of 49 strains of *fosA3*-positive *E. coli*. Strains, sources and PFGE types are indicated, including 13 pulsotypes (A–M) that had similarities >80%.

**Figure 2 antibiotics-10-00777-f002:**
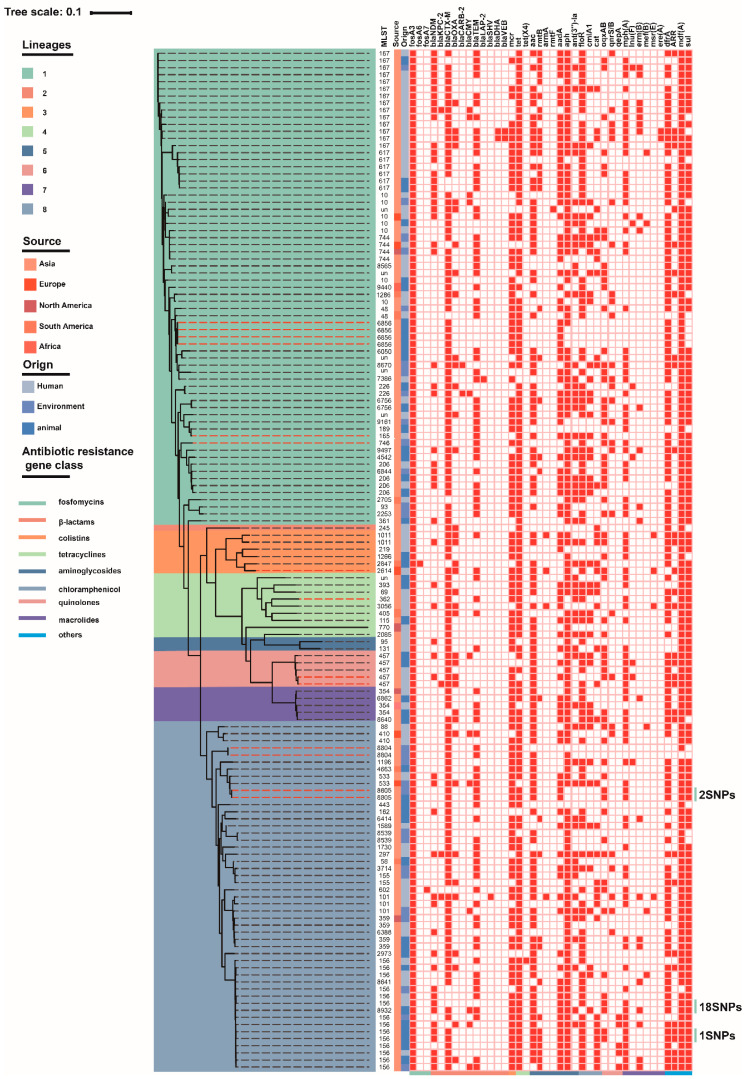
Phylogenetic tree along with ARGs present in the *E. coli* isolates. The phylogenetic tree is composed of 144 *fosA3*-positive *E. coli* divided into eight clusters as indicted. The strains are marked with an orange dashed line in this study.

**Figure 3 antibiotics-10-00777-f003:**
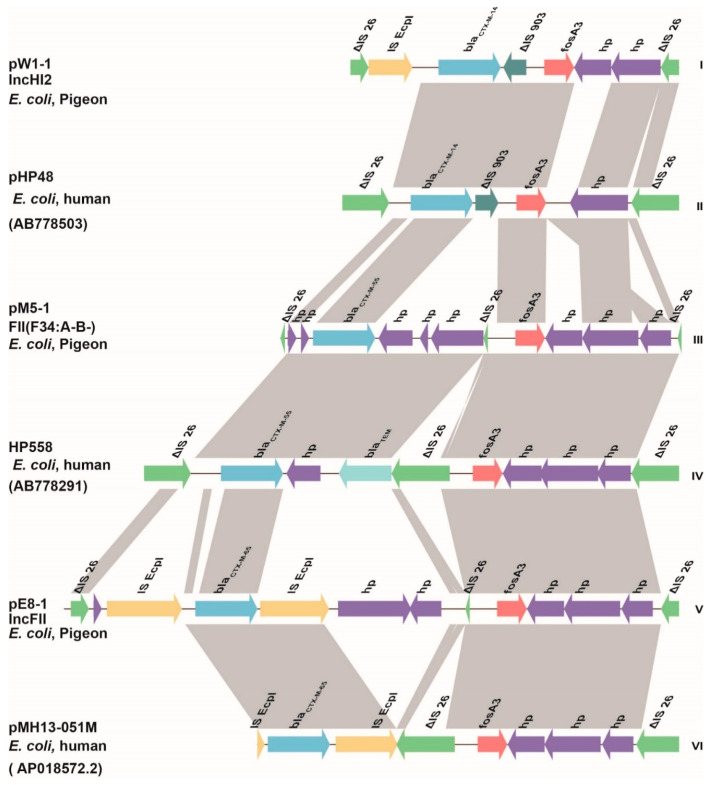
Genetic environment of *fosA3* in 14 *E. coli* isolates. Arrows indicate the directions of transcription of the genes, and different genes are shown in different colours. Regions of >99.0% nucleotide sequence identity are shaded in grey. The delta (Δ) symbol indicates a truncated gene.

## Data Availability

The data for this manuscript is available from correspondence author.

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
