# Peer review of "Molecular Epidemiology of Fosfomycin Resistant E. coli from a Pigeon Farm in China"

_antibiotics, 2021, doi:10.3390/antibiotics10070777_

Round 1

Reviewer 1 Report

In the manuscript ID: antibiotics-1250568, by Han and colleagues, the authors reported the detection and molecular characterization of fosfomycin-resistant, fosA3-positive Escherichia coli strains, isolated from different sources related to a pigeon farm in China. The resistance gene was detected in a high percentage (94.2 %) of E. coli isolates, which were divided in 9 different sequence types and 13 pulsotypes by MLST and PFGE analysis, respectively. The phenotypical and genotypical characterization of fosfomycin resistance evidenced the presence of the fosA3 gene in different plasmids, often in combination with other resistance genes, mostly blaCTX-M; furthermore, the analysis of the genetic context of the gene, in comparison to other deposited genomes, highlighted the presence of shared genetic constructs, thus suggesting a horizontal and clonal spread of this specific fosfomycin resistance determinant.

Although not related to the main topic of the special issue (soil and plant microbiome), the manuscript is very interesting, it presents a wide variety of results and a scrupulous and detailed genetic characterization of the target gene. The methods are well defined and described, the results well presented and commented. There are just few concerns to be addressed before publication of the paper:

-Despite the quality and the abundance of the results, some of them (i.e. the percentages of antibiotic resistant strains or the association with the blaCTX-M gene) could be better justified by statistical analysis;

-The authors stated that the fosA3 gene was spread by horizontal gene transfer and clonal dissemination; although mating experiments were performed, as reported in the material and methods section (lines 80-82), there is no indication of the detected transferability. Could the authors clarify this?

-The authors detected 9 different sequence types and 13 pulsotypes by MLST and PFGE. Could they indicate any correspondence between the obtained results?

Minor revisions are thus recommended to the manuscript before publication in “Antibiotics”.

MINOR COMMENTS

Lines 16 and 18, please use the complete name “Escherichia coli” for the first time the bacterial species is nominated and then the abbreviation “E. coli”;

Line 33, please correct “resistant diseases”;

Lines 34, 35, please correct “there are more than 10 fos types, with fosA and fosC2 being the primary ones found in the Enterobacteriaceae”;

Line 37, please correct “China”;

Lines 38, 39, please correct “…fosA3 gene has been detected in food animals, as well as its transmission in both humans and animals. Thus,…”;

Lines 44, 45, please correct “zoonotic bacteria, spread by faeces”;

Line 48, please correct “global and clonal multidrug-resistant CRE (carbapenem-resistant Enterobacteriaceae) carried by pigeons have the possibility”;

Line 84, please correct “The total DNA of Fosfomycin resistant”;

Line 119, please correct “The 49 fosA3-positive E. coli isolates were resistant to…”;

Line 159, please correct ”The β-lactamase genes displayed the greatest diversity”;

Line 162, please delete “were present” before the percentages;

Line 180, please correct “resistance to fosfomycin has become”;

Please provide a better quality Figure 2, correct “Africa” in the Y axis.

Author Response

Re: Antibiotics-1250568

Dear Editor,

Thanks for your guidance in reviewing our submission. The manuscript has been revised and the editor and reviewers suggestions have been addressed. The reference format has been changed and the language has also been revised. And the yellow highlights are the comments made by the editor and reviewers, as well as the blue highlights are the language changes. We believe the revised manuscript is improved and we hope it will be acceptable for publication.

Thanks for your consideration.

Sincerely yours,

Rong-Min Zhang

Pharmacology & Toxicology

College of Veterinary Medicine

South China Agricultural University,

Guangzhou 510642, China

E-mail: [email protected].

The detailed responses (point-by-point) are listed as follows:

Point 1: Despite the quality and the abundance of the results, some of them (i.e. the percentages of antibiotic resistant strains or the association with the blaCTX-M gene) could be better justified by statistical analysis.

Response 1: Thank you for your suggestion. We have added it in the manuscript. We analyzed the NCBI database of fosA3-positive isolates (downloaded 998 fosA3-positive isolates on November 7, 2020) and WGS (14 fosA3-positive isolates with WGS in this study) found that the positive rate of blaCTX-M was 92.59% (937/ 1012). The positive rates for blaCTX-M-14, blaCTX-M-15, blaCTX-M-55, blaCTX-M-65 were 27.17% (275/1012), 4.94% (50/1012), 36.67% (371/ 1012), 17.29% (175/1012), respectively (lines 180-184).

Point 2: The authors stated that the fosA3 gene was spread by horizontal gene transfer and clonal dissemination; although mating experiments were performed, as reported in the material and methods section (lines 80-82), there is no indication of the detected transferability. Could the authors clarify this?

Response 2: Thank you for your advice. Conjugation experiments were performed on 49 fosA3-positive E. coli isolates collected from pigeon farm and the surrounding environment. Thirty-one fosA3 plasmids were successfully transferred to recipients (E. coli C600str) by conjugation (lines 129-130).

Point 3: The authors detected 9 different sequence types and 13 pulsotypes by MLST and PFGE. Could they indicate any correspondence between the obtained results?

Response 3: Thank you for your comment. The correspondence between MLST and PFGE pulsotypes have been added to Figure 1. Compared with MLST, H-J pulsotypes belonging to ST6856 we also found K-M pulsotypes belonging to ST457. Other PFGE pulsotypes corresponded to MLST (lines 135-136).

Point 4: Lines 16 and 18, please use the complete name “Escherichia coli” for the first time the bacterial species is nominated and then the abbreviation “E. coli”;

Response 4: Thank you for your suggestion and have corrected the manuscript (lines 16-17).

Point 5: Line 33, please correct “resistant diseases”;

Response 5: Thank you for your advice. This was corrected in the manuscript (line 33).

Point 6: Lines 34, 35, please correct “there are more than 10 fos types, with fosA and fosC2 being the primary ones found in the Enterobacteriaceae”;

Response 6: Thank you for your comment. This was corrected in the manuscript (lines 34-35).

Point 7: Line 37, please correct “China”;

Response 7: Thank you for your suggestion. This was corrected in the manuscript (line 38).

Point 8: Lines 38, 39, please correct “…fosA3 gene has been detected in food animals, as well as its transmission in both humans and animals. Thus,…”;

Response 8: Thank you for your advice. This was corrected in the manuscript (line 38).

Point 9: Lines 44, 45, please correct “zoonotic bacteria, spread by faeces”;

Response 9: Thank you for your comment. This was corrected in the manuscript (lines 42-43).

Point 10: Line 48, please correct “global and clonal multidrug-resistant CRE (carbapenem-resistant Enterobacteriaceae) carried by pigeons have the possibility”;

Response 10: Thank you for your suggestion. This was corrected in the manuscript (lines 45-46).

Point 11: Line 84, please correct “The total DNA of Fosfomycin resistant”;

Response 11: Thank you for your advice. This was corrected in the manuscript (line 82).

Point 12: Line 119, please correct “The 49 fosA3-positive E. coli isolates were resistant to…”;

Response 12: Thank you for your comment. This was corrected in the manuscript (line 125).

Point 13: Line 159, please correct “The β-lactamase genes displayed the greatest diversity”;

Response 13: Thanks. This was corrected in the manuscript (lines 167-168).

Point 14: Line 162, please delete “were present” before the percentages;

Response 14: Thank you for your suggestion. This was deleted in the manuscript (line 170-171).

Point 15: Line 180, please correct “resistance to fosfomycin has become”;

Response 15: Thank you for your advice. This was corrected in the manuscript (line 192).

Point 16: Please provide a better quality Figure 2, correct “Africa” in the Y axis.

Response 16: Thank you for your comment. This was corrected in Figure 2 and replaced with a clearer one.

Reviewer 2 Report

The manuscript by Han et al. described the presence of fosfomycin-resistant E. coli in the pigeon farm in China.  The authors concluded the high prevalence of fosfomycin resistant E. coli is a threat for humans. However, the authors selected fosfomycin resistant E. coli by plating the samples on fosfomycin-containing agar at a very high concentration (256mg/L), which is highly possible that the isolates are fosfomycin resistant. Therefore, it is not a proper way to conclude the high prevalence of fosfomycin-resistant E. coli in the pigeon population in China. Further, please give a reason for why the authors used 256mg/L of fosfomycin in the agar when they say MIC >= 4mg/L as "high-level" fosfomycin resistance" in the abstract. Please clarify which criteria was used to define fosfomycin resistant E. coli

Other comments:

Methods
Line 70: Please describe antimicrobial susceptibility testing methods in more detail (i.e., what are the breakpoints for each antibiotic? Which breakpoints were used for which antibiotics (CLSI or EUCAST)?

Were there other resistant genes other than fosA, fosA3, and fosC2 based on WGS analysis?

Line 87: What is the read length of paired-end sequencing?
Line 92: Please describe the selection criteria for WGS information from public resources (fosA3 gene positive).

Line 104: Ethicl - Ethical

Figure 2: Please indicate the isolates from this study.
Figure 2: What are the legends with "Antibiotic resistance gene class" used for? If this is for the line colors, I only see orange lines.
Figure 2: Based on what did the authors formed 8 clusters? In cluster 4, isolates 770 and 2085 seem more like cluster 5. Please show the bootstrap values.

How can the authors conclude that fosfomycin-resistant isolates disseminated from a human population to pigeon from this study?  There is a possibility that the isolates came from pigeons or other animals. It is hard to make such a conclusion from this study without longitudinal studies.

URL in parentheses should be in references.

Author Response

Re: Antibiotics-1250568

Dear Editor,

Thanks for your guidance in reviewing our submission. The manuscript has been revised and the editor and reviewers suggestions have been addressed. The reference format has been changed and the language has also been revised. And the yellow highlights are the comments made by the editor and reviewers, as well as the blue highlights are the language changes. We believe the revised manuscript is improved and we hope it will be acceptable for publication.

Thanks for your consideration.

Sincerely yours,

Rong-Min Zhang

Pharmacology & Toxicology

College of Veterinary Medicine

South China Agricultural University,

Guangzhou 510642, China

E-mail: [email protected].

The detailed responses (point-by-point) are listed as follows:

Point 1: The manuscript by Han et al. described the presence of fosfomycin-resistant E. coli in the pigeon farm in China. The authors concluded the high prevalence of fosfomycin resistant E. coli is a threat for humans. However, the authors selected fosfomycin resistant E. coli by plating the samples on fosfomycin-containing agar at a very high concentration (256 mg/L), which is highly possible that the isolates are fosfomycin resistant. Therefore, it is not a proper way to conclude the high prevalence of fosfomycin-resistant E. coli in the pigeon population in China. Further, please give a reason for why the authors used 256 mg/L of fosfomycin in the agar when they say MIC >= 4mg/L as "high-level" fosfomycin resistance" in the abstract. Please clarify which criteria was used to define fosfomycin resistant E. coli.

Response 1: Thank you for your advice. We have checked the breakpoint according to the CLSI documents M100-S28 again and found that MIC breakpoints >= 256 mg/L as fosfomycin resistance. In this study, we used 256 mg/L as the added concentration of fosfomycin-containing agar. The MIC >= 4 mg/L in the article abstract has been modified to 256 mg/L in the manuscript (line 19).

Point 2: Line 70: Please describe antimicrobial susceptibility testing methods in more detail (i.e., what are the breakpoints for each antibiotic? Which breakpoints were used for which antibiotics (CLSI or EUCAST)?

Response 2: Thank you for your suggestion. MICs were interpreted by referring to standards from CLSI documents M100-S28, and the breakpoints of colistin and tigecycline for Enterobacteriaceae were interpreted according to EUCAST criteria (Version7.0). MIC breakpoints of colistin and tigecycline for Enterobacteriaceae was absent in the CLSI documents M100-S28 (line 70-71).

Point 3: Were there other resistant genes other than fosA, fosA3, and fosC2 based on WGS analysis?

Response 3: Thank you for your comment. We have added it in the manuscript (lines 168-171).

Point 4: Line 87: What is the read length of paired-end sequencing?

Response 4: Thank you for your advice. We have added it in the manuscript and the length of paired-end sequencing was 250 bp (line 84-85).

Point 5: Line 92: Please describe the selection criteria for WGS information from public resources (fosA3 gene positive).

Response 5: Thank you for your advice. We selected these representative 130 isolates according to the following standard. Briefly, the WGS of a total of 774,435 isolates were downloaded from GenBank as of November 7, 2020 and the fosA3-positive E. coli were screened using stand-alone BLAST analysis, resulting in a total of 915 fosA3-positive E. coli. Phylogenetic analysis was performed using the isolates from current study and public repository. Since the phylogenetic software could only process 500 isolates at a time, therefore these 915 fosA3-positive E. coli strains were randomly divided into two groups and each group were used to construct a phylogenetic tree with the 14 isolates in current study. Then the 130 representative strains were selected according to their distribution in the phylogenetic tree, source, and sampling location. These 130 representative strains are marked with blue in Figure S2 (Lines 91-99).

Point 6: Line 104: Ethicl – Ethical

Response 6: Thank you for your suggestion. We have corrected the manuscript (line 109).

Point 7: Figure 2: Please indicate the isolates from this study.

Response 7: Thank you for your advice. The isolates have been marked in the legend in this study.

Point 8: Figure 2: What are the legends with "Antibiotic resistance gene class" used for? If this is for the line colors, I only see orange lines.

Response 8: Thank you for your comment. The "Antibiotic resistance gene class" is for the line colors at the bottom of Figure 2 and are now in bold type (lines 168-169).

Point 9:  Figure 2: Based on what did the authors formed 8 clusters? In cluster 4, isolates 770 and 2085 seem more like cluster 5. Please show the bootstrap values.

Response 9: Thank you for your suggestion. Hierbaps was used for phylogenetic tree cluster. The isolates 770 and 2085 bootstrap values were 87%, 98%, but the Node labels were 0.5907, 0.4557.

Point 10: How can the authors conclude that fosfomycin-resistant isolates disseminated from a human population to pigeon from this study? There is a possibility that the isolates came from pigeons or other animals. It is hard to make such a conclusion from this study without longitudinal studies.

Response 10: Thank you for your advice. We have corrected the manuscript (lines 216-219).

Round 2

Reviewer 2 Report

The authors responded to raised questions appropriately. A few points to improve the manuscript are listed below.

Other points

Line 23: the word "horizontal transmission" is used more for virus or in broader content. Horizontal transfer of plasmid might be more appropriate.

Line 33: "resistant diseases" should be "resistant bacteria"

Line 38: Please add reference.

Line 41: The reference 6 is the data from Spain. This is not appropriate when authors are describing pigeon farm in China. Please refer to the data from China.

Line 66, 87, 101: Please add the web page properly in the reference list. 

Line 78: Please describe in more detail or add reference about filter mating method. 

Author Response

Point 1: Line 23: the word "horizontal transmission" is used more for virus or in broader content. Horizontal transfer of plasmid might be more appropriate.

Response 1: Thank you for your advice and have modified it in the manuscript (lines 23-24, 195, 214).

Point 2: Line 33: "resistant diseases" should be "resistant bacteria".

Response 2: Thank you for your suggestion. This was modified in the manuscript (line 33).

Point 3: Line 38: Please add reference.

Response 3: Thank you for your comment. We have added it in the manuscript (line 38, 234-236).

Point 4: Line 41: The reference 6 is the data from Spain. This is not appropriate when authors are describing pigeon farm in China. Please refer to the data from China.

Response 4: Thank you for your advice. We have replaced the reference in the manuscript (line 42, 239-241).

Point 5: Line 66, 87, 101: Please add the web page properly in the reference list.

Response 5: Thank you for your advice. This was added in the manuscript (line 65, 272-273), (line 86, 286-287), (line100, 294-295).

Point 6: Line 78: Please describe in more detail or add reference about filter mating method.

Response 6: Thank you for your suggestion. We have added a reference about detail filter mating method in the manuscript (lines 78-79).